# Comparing the growth and yield performance of six different varieties of frafra potato (Solenostemon rotundifluis Poir) grown under rain-fed conditions in the Guinea Savanna ecological zone of Ghana

**Abonuusum Ayimbire**[1]*, **Joseph Kunansua Laary**[1], **James Anaba Akolgo**[1], **Augustus Dery Ninfaa**[1], **Joseph Asampana Akolgo**[2], **Abdul Ganiu Anyagri Ndeogo**[1]

**1** Department of Ecological Agriculture, School of Agriculture, Bolgatanga Technical University (BTU), Bolgatanga, Ghana, **2** Northern Development Authority, Lamashegu, Tamale, Ghana

* aayimbire@bolgatu.edu.gh

## 1. Introduction

Frafra potato (*Solenostemon rotundifolius*) is a root tuber crop belonging to the Lamiaceae family and is widely cultivated across tropical African and Asian countries [1,2]. *S. rotundifolius* is known by multiple names across these African and Asian countries, making it very difficult to collect reliable data and harmonize the existing literature on the crop [2]. In Ghana, the crop is mainly grown for food for personal consumption and commercial purposes, and its production is restricted to the Guinea and Sudanian Savanna vegetation zones, covering the regions of Northern Ghana [1,3], bordering Togo and Burkina Faso. It thrives in the dry savanna areas partly because it can withstand adverse weather conditions and other consequences of climatic change, including drought and high temperatures. It also does not require a lot of inputs in order to produce root tubers; thus, farmers can spend their time and resources on other crops. *S. rotundifolius*, whose tubers mature within 4–6 months, can be harvested at leisure after maturity. The plant completes its life cycle within a year [4,5].

*S. rotundifolius* has the potential to grow to a height of about 15 cm, which can increase to 60 cm under favorable conditions. It produces very small, pink, white, pale violet, or blue hermaphrodite flowers, arranged in a raceme-like inflorescence [2]. Though the crop is grown purposely for its underground tuberous roots, the vegetative parts can be processed as ruminant feed, especially during the dry season. It produces ovoid tubers which are rich in nutrients, minerals, vitamins, and phytochemicals [6–8]. The tubers can be left on the farm in dry conditions for about a year without spoilage, especially in the dry season. The crop has some medicinal properties, hence it is named '*Coleus dysentericus*' in some areas, suggesting that it relieves patients of dysentery [1,2,5,9].

These qualities reinforce the place of frafra potato in the category of crops that can be produced by an average subsistence farmer and can contribute to household food security [1,5,6,8].

Even though frafra potato is considered economically viable and nutritionally fortified for rural households, to the best of our knowledge, local farmers and the general public have not been appropriately presented with information on the crop's potential. Consequently, the crop has received little research focus. Hence, its cultivation is restricted to indigenous rural communities and is almost on the verge of extinction [6–8]. Besides, improved varieties are either uncommon or unknown to the local farmers [1,3,6]. Again, people generally are against frafra

**Data Availability Statement:** All relevant data are within the paper and its Supporting Information files.

**Funding:** The author(s) received no specific funding for this work.

**Competing interests:** The authors have declared that no competing interests exist.

potato production and consumption using factors of convenience (for example, that the tuber size is too small) rather than the factors of necessity (for example, nutrient fortification), to neglect the crop which is contributing to its production decline [1].

Therefore, research should be focused on improving tuber size, tuber yield, and vegetative components of the crop using available varieties. Moreover, selecting promising varieties with desirable characteristics in vegetative growth and tuber development can draw the attention of local farmers and the general public to the potential of the crop.

Conducting studies on the existing varieties of the crop under similar growth conditions can help in comparing and identifying high-yielding varieties. This can improve the incomes of crop producers and enhance household food security while safeguarding the crop against extinction [1,8]. This study, therefore, aimed at evaluating the vegetative growth and root tuber yield performance of six frafra potato varieties under rain-fed conditions in the Upper East Region of Ghana.

## 2. Methodology

### 2.1 Study location

The study was conducted in the experimental field of the Department of Ecological Agriculture, Bolgatanga Technical University (BTU), Sumbrungu, located at 10.8296101˚N, 0.9414592˚W and within the Bolgatanga Municipality (10.7875˚N, 0.8580˚W) in the Upper East Region of Ghana (Fig 1).

Bolgatanga Municipal is located within Bolgatanga, the capital of the Upper East Region of Ghana. The region is a part of the Sudanian Savanna ecological zone of Ghana, with erratic rainfall distribution and pattern, averaging 800 mm from May to June and about 1,100 mm from September to October. The average afternoon and night temperatures are about 35˚C and 14˚C, respectively, with low relative humidity. The vegetation generally consists of short, widely dispersed deciduous trees and grasses which have adapted to the low moisture conditions during the long dry seasons [10].

### 2.2 Soil sampling and testing

The study was conducted in the rainy season under conditions similar to those faced by the local farmers for a better assessment of the growth and tuber production and to provide subsequent recommendations to the farmers. Soils were sampled on two experimental sites adjacent to each other at 20 cm depth and analyzed for some properties before planting and after harvesting to assess the soil capacity to support frafra potato crop production following Taylor and Francis [11] and Fery *et al.* [12].

Using a hand probe, three (3) soil samples (0–20cm depth) were taken at equal intervals from the top of each of the four (4) replicate ridges in each experimental site. Collected soil samples were kept in clean, black plastic bags. Thus, there were 12 soil samples per replicate (three samples per ridge by four replicate ridges) and 72 soil samples. Each set of 12 soil samples from each replicate was poured into a clean bucket, well-mixed, parceled into a composite sample of about 30 g, appropriately labeled, and analyzed at the Crop and Soil Sciences laboratory of Kwame Nkrumah University of Science and Technology, Kumasi, Ghana. The soil samples were also taken after harvest using the same procedure and analyzed at the same laboratory.

### 2.3 Land preparation, field layout, and planting

The experiment was conducted in the rainy season from May-November in 2020 and 2021 using six different frafra potato varieties in a randomized complete Block Design (RCBD) with

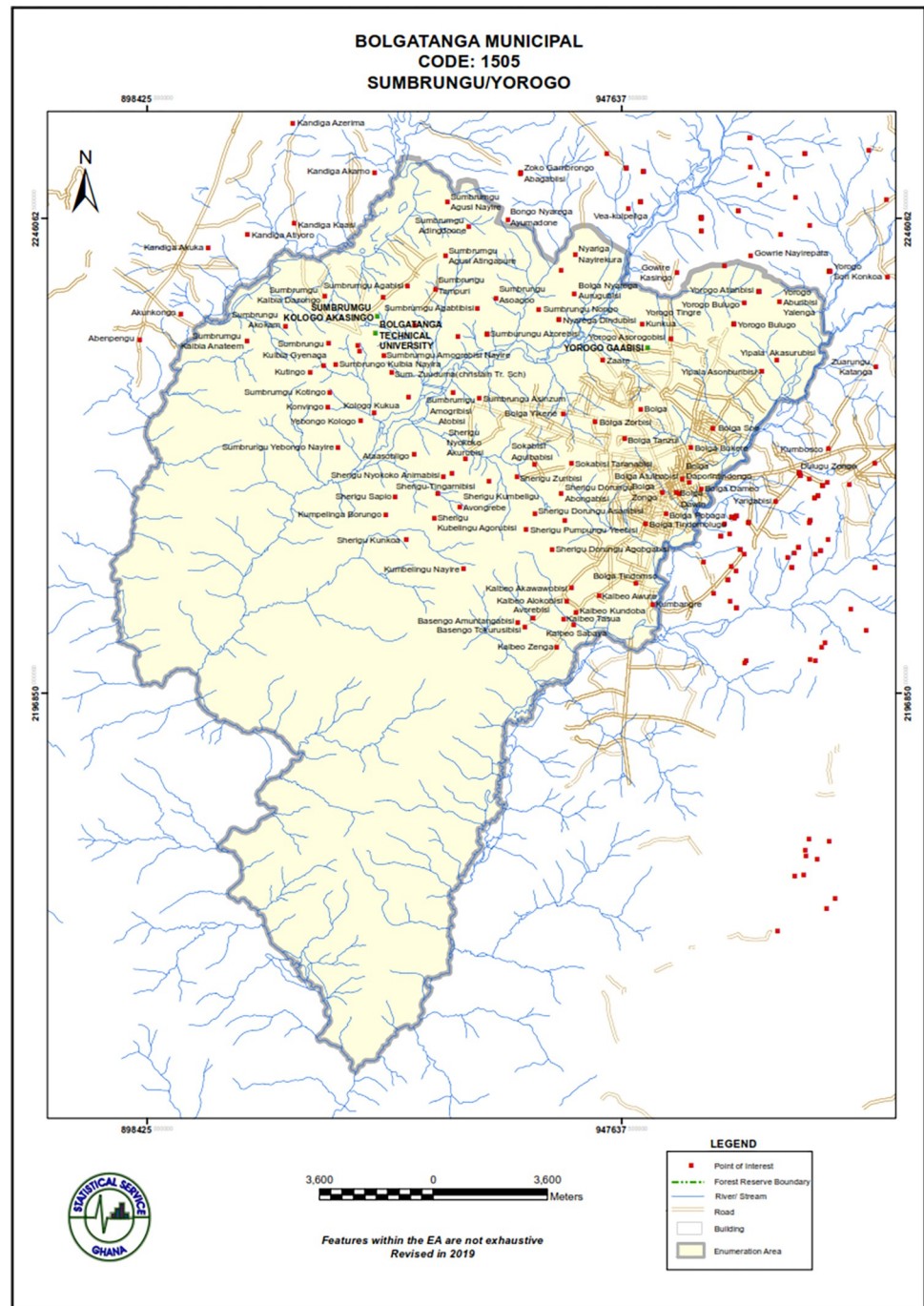

**Fig 1. A map of bolgatanga municipal showing Bolgatanga Technical University located in Sumbrungu Kologo Akasingo.** Source: Ghana Statistical Service.

four replications. These six varieties were Manga-Moya (MM), Maa-Lana (ML), Nutsuga Peisa (NP), Nachim–Tiir (NT), WAAP Peisa (WP) and Local Variety (LV).

In both the rainy seasons, adjacent fields were used for the experiment and were cleared of debris and ridges using a hoe to soften the land before planting. Each ridge was 0.5 m high, 0.7 m wide, and 3 m long. The holes were created with a hoe and sprouted tubers were carefully

planted at the onset of rains in May 2020 and 2021. Sprouted seedlings were gently covered with topsoil to avoid breaking off the sprouted delicate shoot or burying them in the soil. Ridges are suitable for tuberous root formation [13], and tuber yield is likely to be maximized when the sprouted tubers are planted between May and the middle of June [1].

## 2.4 Cultural practices and pest management

The experiment was conducted in the rainy seasons of 2020 and 2021, but plants were sparingly watered in the beginning prior to the onset of rains. The field was weeded thrice after planting. During weeding, the soil was raised with hoe to support the base of the plants and to prepare the plants for shooting pods for tuber formation. Weeds were also occasionally pulled out by hand.

The foliage of the plants was regularly inspected for incidence of pests and diseases using magnifying glasses. The insect pests, such as aphids, white flies, grasshoppers, and caterpillar-like larval forms were occasionally found and identified on the vegetative parts of the frafra potato crop. However, no pesticide was sprayed on the crops because the pests were not at threshold levels to cause appreciable damage or alter the growth and yield performance of the crop. Farmers generally do not consider pests as a threat to frafra potato crop because their leaves are usually not the desired choice of most herbivores [1].

## 2.5 Measurement of growth parameters

Data collection, which was done fortnightly, started a month (about 4 weeks) after planting seeds in both the rainy seasons. Four plants were randomly selected per variety (one plant per replication) for data measurement in each experiment. Thus, data on growth parameters were collected from 24 plants in each experiment. Each selected plant was marked by loosely tying a colored plastic band to one of its branches for easy identification. The following growth parameters were measured: plant height (cm), leaf area ($cm^2$), number of vines, and canopy spread (cm). The mass of tubers (kg) produced by each variety of frafra potato was also measured.

## 2.6 Estimation of leaf area $[cm^2]$

To estimate the leaf area, a leaf was taken out from the randomly selected plants per variety. The selected leaf was labeled by tying it loosely with a colored plastic band. In total, four (4) leaves were labeled per treatment—a total of 24 leaves for the six treatments.

Without plucking the selected leaf from the plant, the leaf area was measured by tracing the outline of the selected leaf blade on a graph sheet using a pencil (Plate 1). The leaf area was estimated by counting every 1 $cm^2$ such that the leaf outline was covered. Suppose the number of 1 $cm^2$ fully covered by the leaf outline was counted to be $x_1$. Also, where at least three-quarters of the squares were covered by the leaf outline, they were counted as being fully covered, which is $x_2$. Then, 1 cm squares that were half-covered by the outline were counted to be $y$, which was divided by two to obtain a full 1 cm square, that is $y/2$. Also, where at most a quarter of squares were covered by the leaf outline, were not counted. Therefore, the total estimated area of the leaf ($cm^2$) was obtained by adding the number of 1 cm squares that were fully covered by the leaf outline ($X_1 + X_2 + Y/2$).

## 2.7 Vine branches and canopy spread [cm]

The number of vine branches produced by the selected plants per variety was manually counted at regular intervals of days (fortnightly) and recorded. The vine branches make up the

canopy, and the canopy spread was determined using a centimeter ruler to measure the radius of the crown of the selected plants. This was obtained by taking the diameter of the crown in two different directions and the average values recorded. This average diameter was then divided by two to obtain the radius of the canopy spread (cm).

### 2.8 Days to flowering and maturity

Days to flowering were determined by observing and recording the dates on which flowering began. The number of days from planting to flowering was determined by comparing the planting date with the date of flower initiation. The days to maturity were also obtained by comparing the date of planting with the date of tuber maturity. The tubers were harvested after maturity.

### 2.9 Tuber mass [kg]

Frafra potato tubers were harvested at maturity on a per plant per variety basis. Tubers of each variety per replicate ridge were harvested separately and tubers were weighed (in Kg) using a beam balance. Thus, the mass of the tubers of each variety per experiment was obtained by summing the mass of tubers of that variety in the replicates and averaging.

### 2.10 Rainfall data

Rainfall data within the Bolgatanga Municipality, covering the period the frafra potato was cultivated, from April to November 2020 and 2021, was obtained from the Ghana Meteorological Agency, Bolgatanga, Upper East Region. Though the rainfall distribution differed in both the seasons, the rainfall amounts were without statistical differences and the average values were used.

### 2.11 Data analysis

The data sets were analyzed using a statistical programme, Stata 16.0, StataCorp LLC, College Station, Texas 77845, USA. Comparison of frafra potato varieties with respect to any measured parameter was at the 5% level of confidence. The analyzed data sets in both the experiments were not significantly different and were combined and analyzed to obtain the results.

## 3. Results and discussion

### 3.1 Soil properties of the study site

Soil analyses (Table 1) show that the soil texture of the study site is mainly sandy loamy, with the proportion of sand being 84.7% before planting which decreased to 84.0% after harvesting. The composition of silt decreased from 6.4% before planting to 5.7% after harvesting. On the contrary, the percentage of clay increased from 8.9% to 10.3% after harvesting.

   The fraction of silt (1.60%) remained about the same before planting and after harvesting. The higher proportion of sand makes the soil particles loose, less adhesive and less compacting, making the soil suitable for root penetration and tuber formation. The soil with a small proportion of silt and clay relative to sand are usually well-draining, which is a good requirement of frafra potatoes for maximum tuber yield [1], despite its low moisture retention capacity.

   The soil was also found to be slightly acidic with an average pH value of 6.6 before planting, decreasing to a pH value of 5.8 after harvesting. Before planting, the soil exchangeable acidity ($Al^{3+}$ and $H^+$ concentrations) was 0.59 cmol/kg of $Al^{3+}$ and 0.32 cmol/kg $H^+$. Whereas, after harvesting, the $H^+$ concentration decreased to 0.23 cmol/kg, and the $Al^{3+}$ remained unchanged, i.e. 0.59 cmol/kg (Table 1). This implies that the frafra potato crop could have facilitated the removal of $H^+$ ions from the soil and may perform better in acidic soils. In the

**Table 1. Soil structure, chemical reaction, and nutrient content of the study site.**

| | Soil sample mean values | |
| --- | --- | --- |
| Soil parameter | Before planting | After planting |
| **Soil structure** | | |
| Sand, % | 84.70 | 84.0 |
| Silt, % | 8.94 | 10.3 |
| Clay, % | 6.36 | 5.74 |
| Texture class | Loamy sand | Loamy sand |
| | | |
| **Organic components** | 0.34 | 0.33 |
| Organic Carbon, % | 0.59 | 0.57 |
| Organic matter, % | | |
| % Total N | 0.13 | 0.12 |
| Avail. P, mg/kg | 21.77 | 17.51 |
| pH | 6.62 | 5.77 |
| **Exchangeable acidity** | | |
| Al, cmol/kg | 0.59 | 0.59 |
| H, cmol/kg | 0.32 | 0.23 |
| | | |
| **Exchangeable cations** | | |
| K, cmol/kg | 0.65 | 0.31 |
| Ca, cmol/kg | 3.2 | 2.98 |
| Mg, cmol/kg | 1.65 | 1.03 |
| Na, cmol/kg | 0.05 | 0.03 |
| | | |
| **Micronutrients** | | |
| Fe, mg/kg | 10.86 | 17.13 |
| Zn, mg/kg | 1.27 | 1.45 |
| Mn, mg/kg | 15.40 | 21.16 |
| Cu, mg/kg | 0.69 | 1.42 |

case of the sweet potato crop, it facilitated the release of $H^+$ ions into the soil and increased soil pH [14].

Soil pH and exchangeable acidity determine the solubility and availability of certain nutrients for plants to use because some nutrients are soluble and available in acidic soils while others dissolve and are available in alkaline or basic soils [15].

The contents of organic matter and organic carbon were low in the study soil before and after the harvest of crop varieties. Among the trace elements tested, the manganese level was 15.4 mg/kg before planting and increased to 21.2 mg/kg after harvesting.

The total nitrogen (N) content before planting was 0.13% and after harvesting, it was 0.12%, suggesting that frafra potato crop may sparingly utilize soil nitrogen for its growth and tuber development, and can grow well in nitrogen deficient soils. On the contrary, the available phosphorus content in the soil before planting was 21.8%, but decreased to 17.5% after harvesting, suggesting that the frafra potato makes more use of soil phosphorus than nitrogen, for its growth and development.

The soil in the experimental site had a limited concentration of nutrients and may require fertilizers or nutrient augmentation to optimize crop yield on the site. Though frafra potatoes may sparingly mine soil nutrients for their growth and development, the crop requires an adequate amount of soil nutrients, especially phosphorus, for its production. Thus, the soil nutrient status of fields must be improved to maximize tuber yield.

**Table 2. Mean values of the growth and yield parameters of frafra potato varieties.**

| Variety | Plant height (cm) | Leaf Area (cm$^2$) | Canopy spread (cm) | Number of vines | Days to Flowering | Days to Maturity | Tuber yield (ton/h) |
|---|---|---|---|---|---|---|---|
| | Mean ± SD | Mean ± SD | Mean ± SD | Mean ±SD | Mean±SD | Mean±SD | Mean ±SD |
| LV | 25.0 ±7.1[a] | 21.2±7.0[b] | 25.0±7.9[a] | 58.8±32.5[a] | 90±5.0b | 138±1.3a | 27.8±4.7[c] |
| ML | 25.4±6.8[a] | 18.3±7.7[a] | 27.6±8.6[a] | 66.0±33.1[b] | 85±5.1a | 138±1.2a | 19.3±2.7[b] |
| MM | 29±10.1[b] | 17.4±5.8[a] | 30.6±11.1[b] | 82.8±46.4[b] | 92±5.9b | 145±1.4a | 15.1±6.4[a] |
| NP | 25.9±7.4[a] | 18.7±6[a] | 28.0±9.9[a] | 63.1±32.8[a] | 93±2.7b | 145±1.3a | 20.8±6.3[b] |
| NT | 26.4±8.8[a] | 20.8±7.3[b] | 26.6±9.4[a] | 56.8±33.1[a] | 84±4.7a | 138±1.1a | 25.0±3.7[c] |
| WP | 31.7±13[b] | 20.3±6.6[b] | 29.6±9.2[b] | 88.2±56.7[c] | 89±7.4b | 145±1.3a | 16.6±3.5[a] |
| Mean | 27.5±8.9 | 19.6±6.8 | 27.9±9.4 | 69.5±38.7 | 89±5.6 | 142±1.4 | 20.7±5.4 |
| %CV | 32.6 | 34.6 | 33.4 | 56.4 | 5.6 | 0.9 | 27.4 |
| P-Value | 0.27 | 0.30 | 0.93 | 0.15 | 0.00 | - | 0.62 |

FPV = Frafra Potato Variety; LV = Local variety, ML = Maa-Lana, MM = Manga-Moya, NP = Nutsuga Peisa, NT = Nachim–Tiir, WP = WAAP Peisa; SD = Standard deviation.

Values with the same superscript letter are not significantly different at 5% level of probability.

## 3.2 Growth and development characteristics of frafra potato

The mean values of the measured growth and development parameters are presented in Table 2. Pictures of the growth and development stages of the frafra potato plants are presented in plates 1–5. There were variations in vegetative growth (plant height, leaf area, canopy spread, vine number) and reproductive growth (days to flowering, days to maturity, and tuber production) among the six frafra potato varieties. Plates 1–5 are photographs of the varieties of frafra potatoes showing the various stages of growth from five weeks after planting to maturity.

**3.2.1 Plant height [cm].** The mean plant height across the six varieties was 27.5 ± 13.8 cm. However, plant height varied from the tallest of 31.7 ± 13 cm in WP to the shortest of 25 ± 7cm in LV. The MM and WP varieties were the tallest and were significantly different (P<0.05) in height from the rest of the varieties (Table 2). This indicates that there are variations in plant height (CV = 33%) among the frafra potato varieties.

Table 3A presents the mean correlation of growth and development parameters to each other and to the crop yield. Plant height across varieties was highly positively correlated to leaf area (r = 0.78). The highest correlation of plant height to leaf area was recorded in the LV (r = 0.93) while the lowest (r = 0.73 was recorded in NP (Table 3B). This indicates a strong association between leaf development and plant height in frafra potato and suggests some level of dependency in growth and development among vegetative parts of the crop.

The plant height was found to positively but weakly correlate with tuber yield in four varieties ((LV: r = 0.06; MM: r = 0.21; NP: r = 0.06; WP: r = 0.17) and negatively correlated to tuber yield in two varieties (ML: r = - 0.067; NT: r = - 0.001) (Table 3B). Similar weak correlation trends in tuber yield and leaf area, canopy spread and vine development were found (Table 3B). These correlation values show that there is virtually no association between plant height or vegetative development and tuber yield and that vegetative growth is not a determining factor for tuber development and yield.

Other investigators have also recorded both positive and negative correlations between the length of vines and root tuber yield, even though they worked on varieties of sweet potato. Ochieng [16] recorded a positive link between vine internode length and root yield of sweet potatoes. Ayimbire et al. [14] found a positive relationship between the vine length and mass of root tuber in three different varieties of sweet potato in the range of r = 0.02 and r = 0.19 as well as negative correlations in two others in the range of r = -0.05 and r = -0.03.

**Table 3. a: Mean correlation of growth parameters to each other and to crop yield.** b: Correlation of growth parameters to each other and to yield in the varieties.

| | Plant height (cm) | Leaf area (cm$^2$) | Canopy spread (cm) | Number of vines | Days to flowering | Days to maturity | Yield |
|---|---|---|---|---|---|---|---|
| Plant height (cm) | 1.00 | | | | | | |
| Leaf area (cm$^2$) | 0.78 | 1.00 | | | | | |
| Canopy spread (cm) | 0.95 | 0.81 | 1.00 | | | | |
| Number of vines | 0.93 | 0.70 | 0.93 | 1.00 | | | |
| Days to flowering | -0.07 | 0.01 | -0.02 | -0.02 | 1.00 | | |
| Days to maturity | 0.12 | -0.10 | 0.11 | 0.17 | 0.40 | 1.00 | |
| Yield | 0.05 | -0.01 | 0.02 | -0.02 | -0.77 | -0.21 | 1.00 |

| **LV variety** | Plant height (cm) | Leaf area (cm$^2$) | Canopy spread (cm) | Number of vines | Days to flowering | Yield |
|---|---|---|---|---|---|---|
| Plant height (cm) | 1.00 | | | | | |
| Leaf area (cm$^2$) | 0.93 | 1.00 | | | | |
| Canopy spread (cm) | 0.98 | 0.94 | 1.00 | | | |
| Number of vines | 0.91 | 0.87 | 0.93 | 1.00 | | |
| Days to flowering | -0.05 | -0.06 | -0.05 | -0.11 | 1.00 | |
| Yield | 0.06 | 0.0598 | 0.06 | 0.10 | -0.99 | 1.00 |
| **ML variety** | | | | | | |
| Plant height (cm) | 1.00 | | | | | |
| Leaf area (cm$^2$) | 0.82 | 1.00 | | | | |
| Canopy spread (cm) | 0.95 | 0.82 | 1.00 | | | |
| Number of vines | 0.92 | 0.74 | 0.96 | 1.00 | | |
| Days to flowering | 0.06 | -0.03 | 0.04 | -0.02 | 1.00 | |
| Yield | -0.07 | 0.02 | -0.02 | 0.04 | -0.94 | 1.00 |
| **MM variety** | | | | | | |
| Plant height (cm) | 1.00 | | | | | |
| Leaf area (cm$^2$) | 0.81 | 1.00 | | | | |
| Canopy spread (cm) | 0.96 | 0.87 | 1.00 | | | |
| Number of vines | 0.94 | 0.79 | 0.96 | 1.00 | | |
| Days to flowering | -0.21 | 0.06 | -0.10 | -0.04 | 1.00 | |
| Yield | 0.21 | -0.03 | 0.14 | 0.07 | -0.92 | 1.00 |
| **NP variety** | | | | | | |
| Plant height (cm) | 1.00 | | | | | |
| Leaf area (cm$^2$) | 0.73 | 1.00 | | | | |
| Canopy spread (cm) | 0.96 | 0.78 | 1.00 | | | |
| Number of vines | 0.91 | 0.74 | 0.95 | 1.00 | | |
| Days to flowering | -0.04 | 0.30 | 0.015 | 0.09 | 1.00 | |
| Yield | 0.06 | -0.28 | -0.003 | -0.09 | -0.99 | 1.00 |
| **NT variety** | | | | | | |
| Plant height (cm) | 1.00 | | | | | |
| Leaf area (cm$^2$) | 0.90 | 1.00 | | | | |
| Canopy spread (cm) | 0.98 | 0.89 | 1.00 | | | |
| Number of vines | 0.93 | 0.78 | 0.95 | 1.00 | | |
| Days to flowering | 0.014 | -0.005 | -0.05 | -0.057 | 1.00 | |
| Yield | -0.0005 | 0.03 | 0.027 | 0.0036 | -0.86 | 1.00 |
| **WP variety** | | | | | | |
| Plant height (cm) | 1.00 | | | | | |
| Leaf area (cm$^2$) | 0.78 | 1.00 | | | | |
| Canopy spread (cm) | 0.95 | 0.90 | 1.00 | | | |

(*Continued*)

| | | | | | | |
|---|---|---|---|---|---|---|
| Number of vines | 0.95 | 0.75 | 0.93 | 1.00 | | |
| Days to flowering | -0.21 | 0.19 | -0.12 | -0.14 | 1.00 | |
| Yield | 0.17 | -0.16 | 0.11 | 0.10 | -0.96 | 1.00 |

**3.2.2 Leaf area [cm$^2$] development.** The mean leaf area across varieties was 19.6 ± 6.8 cm$^2$. The highest per leaf area of 21.2±7 cm$^2$ was recorded in LV, while the least per leaf area of 17.4 cm$^2$ was recorded in MM. However, per leaf area values between MM, ML and NP were not significantly different; and that among WP, NT, and LV were also not significantly different at 5% level of confidence. The differences in per leaf area values in the latter three varieties were significantly higher (P< 0.05) than in the former. However, across all varieties, the leaf area was not significantly different (P = 0.30) (Table 2).

In addition, the average standard deviation value of 6.8 across implies that the leaf area is widely spread among frafra potato varieties. Given that leaf area is directly related to water loss through evapotranspiration, the local variety with the largest leaf area might lose more water and will possibly require more water for its growth and development.

However, Table 3A shows that per leaf area of the varieties is negatively correlated to the tuber yield (r = -0.01), which is an indication that tuber development is independent of the leaf area, and leaf size is not a yield determinant in case of frafra potato. It does appear that the leaf area is directly proportional to the vegetative organs of the plants. Therefore, an increase in plant leaf area beyond a certain critical value, depending on the level of growth of the plant, promotes the growth of vegetative parts, by influencing the conduction of photosynthesized food to these parts, leading to an increased vegetative biomass production at the expense of tuber yield [14,17–20].

This may imply that by default, food synthesized in the leaves is more likely to be translocated to the growth of vegetative structures at the expense of tuber formation and development. This probably explains why plants under normal growth conditions tend to gather vegetative biomass from germination or sprouting, and only start to channel assimilates to reproductive parts during stress or the reproductive phase.

**3.2.3 Plant canopy spread [cm].** A mean plant canopy spread of 27.9±9 cm was recorded across all six varieties. These varieties appeared to be similar in stature but varied in canopy spread (CV = 33%). The MM variety showed the widest plant canopy spread of 30.6±11 cm, whereas the narrowest plant canopy spread of 25.1± 8 cm was recorded in LV. The canopy spread in WP and MM varieties was significantly (P< 0.05) higher than in the other varieties (Table 2).

The canopy spread strongly correlated positively with leaf area, with the LV variety showing the highest value (r = 0.94) and the NP variety showing the least value (r = 0.78) (Table 3B) These strong positive relationships suggest some high levels of interdependence between leaf area and canopy spread in frafra potato.

**3.2.4 Vine branches.** The average number of vine branches produced in all the six frafra potato varieties was found to be 70 ± 39. However, the highest number of vine branches per plant (88 ± 57) was recorded in WP and the lowest number (57±33) in NT. Though there were variations in the number of vine branches across the six frafra potato varieties, these differences were not statistically significant (P = 0.15), suggesting that the varieties are inherently similar in vine development (Table 2). There was a high positive correlation (r = 0.70) between leaf area and the number of vine branches formed in all six varieties. The correlation was highest in LV (r = 0.86) and lowest with both ML and NP (r = 0.74) (Table 3A and 3B). This probably portrays the linear relationship between the numbers of vine branches and leaves, given that the more branches the plant has, the more leaves it produces.

Generally, the correlation between vegetative parts like plant height, canopy spread, vine branches, and leaf area is in synchrony with physiology, because the growth of these vegetative parts is dependent on the food synthesized from the leaves. Thus, the larger the leaves, the higher the surface area for trapping both carbon dioxide and sunlight for photosynthesis and the greater is the assimilation availability for vegetative biomass production. Additionally, the growth performance of these vegetative organs of frafra potato could be determined by the available soil water and nutrients, thereby influencing the tuber formation.

**3.2.5 Flowering and maturity periods.** The days to flowering in the six varieties ranged from 84 ± 5 days in NT to 93 ± 3 days in NP, with an overall average of 89 ± 6.0 days (about 3 months) after planting (Table 2). Though days to flowering is an important measure of yield, there was a strong negative correlation (r = - 0.8) between days to flowering and tuber yield in frafra potato, suggesting that early or late days to flowering do not necessarily translate to tuber development. It also indicates that flowering is not a determinant of tuber formation in frafra potato.

In frafra potato, the observable signs of tuber maturity are the yellowing of the leaves, withering and falling of the flowers as well as cracks on the ridges due to expanded tubers [1,5]. All the six varieties developed mature tubers ready for harvest within an average period of 142 ± 1 days (about 5 months) after planting. Three of the varieties (LV, ML, and NT) matured within 138 ±1 days after planting, while the other three varieties (MM, NP and WP) matured within 145±1 days after planting (Table 4).

This maturity period is in line with the results of Kana *et al.* [21] and Enyiukwu *et al.* [2] that frafra potatoes mature within 5–6 months after planting. The number of days to maturity correlated negatively with the tuber yield (r = - 0.21), indicating that tuber yield does not depend on the crop maturity period of the crop. Hence, early or late maturing variety can give either good or poor yield.

**3.2.6 Tuber yield of the studied six varieties of frafra potato.** The mean tuber yield across the six frafra potato varieties was found to be 20.7 ± 5.4 tons per hectare (t/ha) (Table 2). There were significant variations (P = 0.03) in tuber yield across the six varieties, indicating inherent differences of the varieties in tuber production. The highest tuber yield of 27.8 ±4.7 t/ha was recorded in the local variety (LV) and the lowest of 15.1 ± 6.4 t/ha was recorded in Manga-Moya (MM) variety. Though, the LV plants, which gathered the least vegetative biomass, were the shortest plants with the narrowest canopy spread and among those with the lowest number of vine branches. It recorded the highest leaf area and tuber yield. The large leaves probably produced more photosynthates and channeled the same into more tuber formation than vegetative structures.

**Table 4. Mean flowering and maturity periods in the six varieties of frafra potato.**

| Variety | Days to Flowering | Days to Maturity |
|---|---|---|
| | Mean ± SD | Mean ± SD |
| LV | 90±4.95 | 138±0.00 |
| ML | 85±5.11 | 138±0.00 |
| MM | 92±5.89 | 145±0.00 |
| NP | 93±1.69 | 145±0.00 |
| NT | 84±4.70 | 138±0.00 |
| WP | 89±7.42 | 145±0.00 |

FPV = Frafra Potato Variety; LV = Local variety, ML = Maa-Lana, MM = Manga-Moya, NP = Nutsuga Peisa, NT = Nachim–Tiir, WP = WAAP Peisa; SD = Standard deviation.

On the other hand, the MM variety which formed the largest vegetative organs, being the tallest with the highest canopy spread, and number of vine branches produced the smallest leaves and tuber yield. Generally, varieties with high vegetative biomass recorded relatively lower tuber yields (plates 4–6), indicating that the assimilates were channeled into vegetative development at the expense of tuber formation. These observations are supported by the positive correlations (Table 3A and 3B) between yield and leaf area of different plant varieties which have comparatively large leaves but are short and have a small number of vine branches and narrow canopies (LV: r = 0.06; ML: r = 0.02; NT: r = 0.03). On the contrary, the correlation between yield and leaf area of plants in varieties with small leaves which are comparatively tall, have a large number of vine branches and wide canopies are negative (MM: r = -0.03; NP: r = -0.28; WP: r = -0.16). Given that the nutrient content of the soil was found to be very low, containing only 0.13% and 0.12% nitrogen before planting and after harvest (Table 1) and no fertilizers were applied after the soil analysis, the massive vegetative biomass production at the expense of tuber formation could not occur due to excessive nutrients in the soil, especially nitrogen. These findings corroborate the reports that the leaf area is directly related to the growth of vegetative organs or root tubers in the plant [14,17–20].

The tuber yield of the six varieties in this study under the rainfall condition, without the addition of manure or fertilizers is encouraging (plate 6). This was higher than the projected 5–15 t/ha under favorable growth conditions [3], which was an even lower value than the postulated 45 MT/Ha [2,22]. It is, therefore, obvious that a very high yield of frafra potatoes is achievable at the study site under optimum growth conditions.

## 3.3 Relationship between rainfall and leaf expansion in frafra potato

Leaves of the six frafra potato varieties expanded steadily from four (4) weeks after planting (WAP) until 12 WAP (Fig 2A). Except for the Local variety (LV), the leaves of other varieties probably reached their maximum size and stopped expanding 12 WAP. Leaf area of the LV, Manga-Moya (MM), and Maa Lana (ML) increased consecutively from 4–8 WAP (Fig 2A).

Though there was a decrease in rainfall by the 8[th] WAP, and consequent decline in leaf growth; the rate of increase in leaf area within 4–8 WAP was higher than from 10–12 WAP, when rains were rather heavier (Fig 2A). The reduction in leaf expansion within that growth phase could be due to plants approaching maturity, and leaves attaining their maximum sizes or due to excessive soil moisture [2].

The MM variety, however, did not recover from the mild drought and continued at a reduced rate of growth till maturity. The other five varieties recovered and continued to grow, though at varying rates, till maturity at 14 WAP (Fig 2A) and may be considered as being drought-resistant to varying degrees [8,23].

## 3.4 Relationship between rainfall and plant height in frafra potato

The six varieties progressively grew taller with time, from 4–10 WAP (Fig 2B). However, the WAAP Piesa (WP) variety grew vertically fastest as compared to LV and ML varieties (Fig 2B).

Though the rainfall decreased at the 8[th] WAP, it did not appear to have much influence on the vertical growth of the varieties studied. This shows that moderate rainfall could suffice for the growth and development of frafra potato crops. However, there was increased vertical growth especially in WP and Manga-Moya (MM) varieties during heavy rains at the 10[th] WAP.

Plant height of the Nutsuga Piesa (NP), Nachim–Tiir (NT), Maa-Lana (ML), and Local Variety (LV) declined considerably from the 10[th] - 12[th] WAP following the onset of heavy rains and increased slightly as the rains subsided (Fig 2B). The increase in vertical growth due

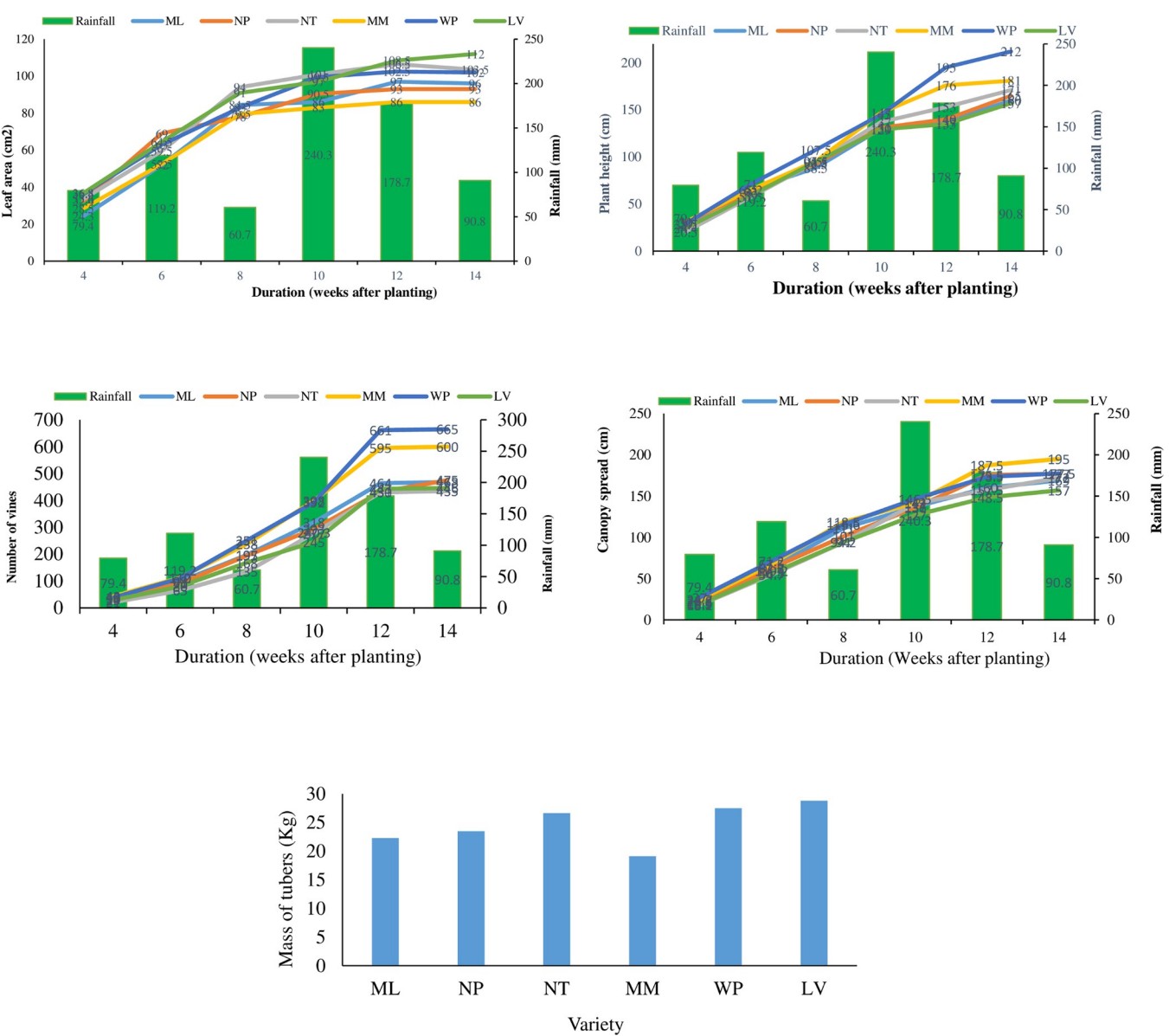

**Fig 2.** a: Relationship between rainfall and leaf expansion in frafra potato. b: Relationship between rainfall and plant height in frafra potato. c: Rainfall and vine production in frafra potato. d: Rainfall and plant canopy spread in frafra potato. e: Tuber mass production in frafra potato. LV = Local variety, ML = Maa-Lana, MM = Manga-Moya, NP = Nutsuga Peisa, NT = Nachim–Tiir, WP = WAAP Peisa.

to moderate rainfall and reduction during excess rainfall is in line with the report that frafra potato crops perform well under moderate rainfall within 700–1,100 mm per annum [2,23].

## 3.5 Relationship between rainfall and vine branching in frafra potato

In all the six varieties, vine branch development increased slowly from 4–6 WAP at varying degrees (Fig 2C). With exception of the NT variety, vine branching of other varieties increased sharply from the 6th WAP till the 10th WAP, when rains were in excess. Similar to their vertical growth, the MM and WP varieties from the 6th WAP increased steadily in vine branching until the 12th WAP (Fig 2C). However, vine branching in the NT variety only increased appreciably

from the 8th WAP to the peak at the 12th WAP. The reduction in rainfall at the 8th WAP did not appreciably affect the branching of vines in all the six varieties (Fig 2C).

Compared to other varieties, the MM and WP produced more vine branches from the 8th– 10th WAP, supporting the earlier reports that these varieties are drought resistant and require moderate rainfall conditions for growth and development [2,8,23]. There was a sharp decline in vine branching from the 12th WAP and could be attributed to the initiation of reproductive growth which requires more assimilates for tuber formation and development, instead of vine production.

### 3.6 Rainfall and plant canopy spread in frafra potato

The canopy spread of the six varieties was found at a steady rate from 4th to the 8th WAP (Fig 2D). However, canopy development in four (4) of the varieties (ML, MM, WP, and LV) were slow in the 8th WAP probably due to limited rainfall. However, the canopy of the NP variety continued to spread from 4th WAP until the 12th WP (Fig 2D).

The rate of growth of the canopy of the NT variety even increased at the 8th WAP, despite the drought, but rather decreased after the 10th WAP when heavy rainfall set in. Thus, except for the MM variety, whose canopy growth rate increased after the heavy rainfall, the latter does not appear to promote canopy expansion in the rest of the varieties of frafra potatoes.

In this study, there was a very massive vegetative development across the six frafra potato varieties. Though the fresh vegetative parts of frafra potato are not a choice for livestock, such massive vegetative parts could be processed into animal feed [2].

### 3.7 Tuber production in frafra potato

Fig 2E shows the mass of tubers produced by each of the six frafra potato varieties. The tubers in the LV variety produced the highest average mass of 28.80 Kg, whereas the lowest tuber mass of 19.15 Kg was recorded in the MM variety.

Generally, there was a massive vegetative biomass production in the plant varieties at the expense of tuber formation, probably attributable to the excessive rainfall during the reproductive phase of the crop, from the 10th WAP. The plants started flowering from 84 ±5 days in NT variety to 93 ± 3 days in NP variety after planting, and were ready for harvest at 138 ± 1 to 145 ±1 days after planting (Table 2).

This shows that though moisture is necessary for tuber development in frafra potato, moderate moisture, particularly at the reproductive phase of the crop promotes better tuber mass and size. This is in line with the report that tuber production in frafra potatoes is highly dependent on the quantity and pattern of rainfall distribution, with moderate and evenly distributed rains favoring good tuber production, while heavy rainfall is counter-productive [2,22,24].

## 4. Conclusion

The six frafra potato varieties produced massive vegetative biomass and a considerable yield of tubers under rainfall conditions. In this study, the LV, NT, and NP varieties performed much better in tuber formation under the rain-fed conditions, and are selective for farmers in rainy season production. Moderate rainfall, especially during the reproductive phase of the crop can promote better tuber mass and size. The LV variety produced the highest tuber yield but recorded the lowest vegetative biomass. The MM and WP varieties produced the highest vegetative biomass but the lowest tuber yield. The NT variety had moderate vegetative structures and tuber yield. This result is an indication that if given the needed attention, the frafra potato crop can produce high tuber yields to augment the food security in households and provide fodder as well for livestock.

## Recommendations

Base on the results, the following recommendations are worth considering:

i. The local variety (LV), Nachim–Tiir (NT) and Nutsuga Peisa (NP) varieties had better tuber yields and are selective for rainy season production

ii. Planting should be timely to escape excessive rains during the tuber initiation and formation stages of the crop

iii. Soil testing and nutrient management should be encouraged at farmer level to improve yields

## Supporting information

**S1 Fig. Plate 1: Tracing the outline of a leaf on a graph to estimate its area.**
(DOCX)

**S2 Fig. Plate 2: The Frafra potatoes five weeks after planting.**
(DOCX)

**S3 Fig. Plate 3: The Frafra potatoes seven weeks after planting.**
(DOCX)

**S4 Fig. Plate 4: The Frafra potatoes three months after planting.**
(DOCX)

**S5 Fig. Plate 5: The Frafra potatoes at maturity.**
(DOCX)

**S6 Fig. Plate 6: Tuber yields in the six varieties of frafra potato.** LV = Local variety, ML = Maa-Lana, MM = Manga-Moya, NP = Nutsuga Peisa, NT = Nachim–Tiir, WP = WAAP Peisa.
(DOCX)

## Acknowledgments

We are grateful to the Savanna Agriculture Research Institute (SARI) for graciously providing us with five of the six frafra potato varieties studied. We are also thankful to the Department of Ecological Agriculture of Bolgatanga Technical University, for granting us the use of their experimental field. We acknowledge the following members of BTU Research Group 2015 who are not authors of this paper: Gabriel Asumboya, Abdul-Rahaman Saibu Salifu, Akazotiyele Richard and Atingah Abi Christina.

## Author Contributions

**Conceptualization:** Abonuusum Ayimbire.

**Data curation:** Joseph Kunansua Laary, Joseph Asampana Akolgo.

**Formal analysis:** Joseph Asampana Akolgo.

**Investigation:** Abonuusum Ayimbire, Joseph Kunansua Laary, Abdul Ganiu Anyagri Ndeogo.

**Resources:** Abonuusum Ayimbire, James Anaba Akolgo, Augustus Dery Ninfaa, Joseph Asampana Akolgo.

**Supervision:** James Anaba Akolgo, Augustus Dery Ninfaa.

**Visualization:** Abonuusum Ayimbire, Joseph Kunansua Laary.

**Writing – original draft:** Abonuusum Ayimbire.

**Writing – review & editing:** Abonuusum Ayimbire, Joseph Kunansua Laary, James Anaba Akolgo.

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
