## [Decision Letter · Decision Letter 0]

25 Jul 2022

PONE-D-22-14687Comparing the Growth and Yield Performance of Six Varieties of Frafra Potato (Solenostemon rotundifluis Poir) Grown under rain-fed conditions in the Guinea Savanna Ecological Zone of GhanaPLOS ONE

Dear Dr. Ayimbire,

Thank you for submitting your manuscript to PLOS ONE. After careful consideration, we feel that it has merit but does not fully meet PLOS ONE’s publication criteria as it currently stands. Therefore, we invite you to submit a revised version of the manuscript that addresses the points raised during the review process.

We look forward to receiving your revised manuscript.

Kind regards,

Min Huang

Academic Editor

PLOS ONE

Journal Requirements:

5. One of the noted authors is a group or consortium BTU Research Group 2015. In addition to naming the author group, please list the individual authors and affiliations within this group in the acknowledgments section of your manuscript. Please also indicate clearly a lead author for this group along with a contact email address.

6. We note that Figure 1 in your submission contain [map/satellite] images which may be copyrighted. All PLOS content is published under the Creative Commons Attribution License (CC BY 4.0), which means that the manuscript, images, and Supporting Information files will be freely available online, and any third party is permitted to access, download, copy, distribute, and use these materials in any way, even commercially, with proper attribution. For these reasons, we cannot publish previously copyrighted maps or satellite images created using proprietary data, such as Google software (Google Maps, Street View, and Earth). For more information, see our copyright guidelines: http://journals.plos.org/plosone/s/licenses-and-copyright.

7. We note that Supplementary Figure plate 4 includes an image of a participant in the study. 

8. Please upload a copy of Figure 3, to which you refer in your text on page 22. If the figure is no longer to be included as part of the submission please remove all reference to it within the text.

9. Please ensure that you refer to Figure 2 in your text as, if accepted, production will need this reference to link the reader to the figure.

Reviewers' comments:

Reviewer's Responses to Questions

**Comments to the Author**

1. Is the manuscript technically sound, and do the data support the conclusions?

Reviewer #1: Partly

Reviewer #2: Yes

2. Has the statistical analysis been performed appropriately and rigorously? 

Reviewer #1: Yes

Reviewer #2: Yes

3. Have the authors made all data underlying the findings in their manuscript fully available?

Reviewer #1: Yes

Reviewer #2: Yes

4. Is the manuscript presented in an intelligible fashion and written in standard English?

Reviewer #1: Yes

Reviewer #2: Yes

5. Review Comments to the Author

Reviewer #1: - Too many Kew words. even though it is determined based on the journal key words number, It must be not more than 7.

- 2 must be supperscribed

- make number 2 'supper scribt'

- Gramatically not correct. b/s single variety require singular verb was. Or the sentence has to be rewritten.

- 'with' must be changed to 'while'.

- you must reason out in the result or duscaution why? leaf area and yield is negetively related b/s it must be + related.

- remove future tense 'will'.

- why? u are measuring the area of the leaf. area means the total coverege of the leaves and u selected one leaf for one replication which is not reponsible leaf nuber to detemine leaf area. B/s a plant has leaf size difference from bottom to top. so that for one plant it need at least three leaf top matured leaf, meduim and botom leaves samplimg. In my opinion the result obtaioned from this sumpling is not have value.

- you should site the tables with results respectively. You should have to have also the results wihich show the significance level in your table following all the parametters you measured and analyized.

Reviewer #2: Major comments

The article can be published according to journal policy. Proof reading is highly recommended to minimize the grammar and punctuation errors/weakness, which is generally conspicuous in write up. But the content and conclusions are acceptable for publication

Minor comments

Abstract:

Line 29: the sentence may end at: deprived soils.

Line 34&38: Punctuate: , respectively

Line 46: I suggest the last sentence should be replaced by the sentence at Line 522-524. But should exclude the part that read ‘’and provide fodder as well for livestock”. In fact, animals hardly feed on the leaves of Frafra potato, due to the off-flavour

Introduction

Line 72: animal feed?

Line: 74-75. The information may not be correct. (sentence can modified or deleted) : the tubers rot if not harvested early under the hot temperature in northern Ghana.

Line 79: These qualities

Line 124: average night temperature of 14 oC. Please check again

Line 169: the sentence may end at: appreciable damage.

Line 408-410: not clear

Line 413: Largest vegetative organ: what does this mean, and where is vegetative organ in Table 3

523-524. Not sure if the leaves are use for livestock feeding, because of the mint flavour

Conclusion

Nil

Recommendation

Nil

Figures.

Figures.2.1 to 2.4: Titles should read: Relationship between rain fall and leaf area expansion , etc

Fig. 5.2 Tuber mass. Not clear to me. I do not understand what this is about.

6. PLOS authors have the option to publish the peer review history of their article (what does this mean?). If published, this will include your full peer review and any attached files.

Reviewer #1: No

Reviewer #2: No

---

## [Author Response · Author response to Decision Letter 0]

4 Oct 2022

To take advantage of our partnership with Editage, visit the Editage website (www.editage.com) 

Manuscript has been edited by Editage. Therefore, I present two copies as follows:

i. Revised manuscript with Track Changes. This was corrected by the authors to effect the reviewers’ comments. 

ii. Manuscript. This manuscript is an unmarked copy (without track changes). This is the revised, editage edited copy which is submitted now to be considered for publication.

Comment of reviewer one and author’s response:

Too many Kew words Thanks. Please, they have been reduced to seven 

2 must be supperscribed Thanks. Please, it has been done (cm2)

Gramatically not correct. b/s single variety require singular verb was. Or the sentence has to be rewritten. Thanks very much, please. However, the statement “Plants of WAAP Peisa were tallest (31.7 cm) as those of Local Variety were shortest (25.0 cm)”, is Grammatically correct because the verb “were” refers to the noun “plants” of WAAP Peisa, in the first part of the sentence which is replaced by the pronoun “those” in the second part of the sentence. Therefore, the use of verb “were” in both parts of the sentence is correct.

'with' must be changed to 'while' Thanks. Please, I have done so

you must reason out in the result or duscaution why? leaf area and yield is negetively related b/s it must be + related Thanks. Please, this has been well explained in the results and discussion section.

why? u are measuring the area of the leaf. area means the total coverege of the leaves and u selected one leaf for one replication which is not reponsible leaf nuber to detemine leaf area. B/s a plant has leaf size difference from bottom to top. so that for one plant it need at least three leaf top matured leaf, meduim and botom leaves samplimg. In my opinion the result obtaioned from this sumpling is not have value.

 Thanks. Please, I agree that using three leaves may cover the various sizes of the leaves at the three stages of growth. However, using one leaf also gives a good estimate because at the begining of measurement, the leaf was young and small and its size was measured. Then, that same leaf grew to maturity and because the measurement was periodical, the various sizes of the leaf at the different stages of growth were measured. Again, each variety had three replicates and a leaf was measured from each replicate which means three leaves were measured from each variety. Given that the comparison was between varieties, the average area of three leaves was actually used, which is in line with your opinion. 

you should site the tables with results respectively. You should have to have also the results wihich show the significance level in your table following all the parametters you measured and analyized. Thanks. Please, these have been done.

Why one-quarter of 1 cm2 was excluded in the estimation of leaf area? Thanks. Please, in the estimation, where the leaf outline covered only three-quarters of the 1 cm2, they were considered as fully covered. This means one-quarter of each of these squares was included in the leaf area even though they were outside the leaf outline. Therefore, where only one-quarter of the 1 cm2 was covered by the leaf outline, this was excluded to balance the other one-quarter of 1 cm2 that was included though it was outside the leaf outline.

Comment of reviewer two and author’s responses:

Proof reading is highly recommended to minimize the grammar and punctuation errors/weakness, which is generally conspicuous in write up Thanks. Please, proof-reading will be done.

Line 29: the sentence may end at: deprived soils Thanks. Please, this has been done,

Line 34&38: Punctuate: , respectively Thanks. Please, I have done so.

Line 46: I suggest the last sentence should be replaced by the sentence at Line 522-524. But should exclude the part that read ‘’and provide fodder as well for livestock”. In fact, animals hardly feed on the leaves of Frafra potato, due to the off-flavour Please, I have done so.

However, the issue of whether animals eat Frafra potato leaves or not, is debatable. Animals, when they have choices to make, will reject some foods. However, in places where the extensive or semi-extensive systems of rearing animals is practiced, and where the dry season covers 7 or 8 months, at the peak of the dry season, animals eat any plant matter they come across. Therefore, if they are served with Frafra potato leaves, they eat voraciously. 

Line 72: animal feed? Please, I have explained this above. In the dry season, animals in my area eat Frafra potato leaves voraciously.

Line: 74-75. The information may not be correct. (sentence can modified or deleted) : the tubers rot if not harvested early under the hot temperature in northern Ghana Thanks. Please, this is a finding by Sugri et al. (2013). They found out that some farmers did so. Living in northern Ghana myself, I can confirm that when the rains stop in the dry season, there is no adequate moisture in the soil to sustain microbial activities to cause rotting! What I have experienced in my area is that the tubers lose water and shrink in sizes. The soil dries up and locks the tubers in the soil. Therefore, depending on the nature of the soil, you may find it difficult to dig and harvest the tubers if they are left in the field for a long time in the dry season. However, if it rains in the dry season, then the tubers immediately start rotting. 

Line 79: These qualities Thanks. Please, I have corrected it

Line 124: average night temperature of 14 oC. Please check again Thanks. Please, this is from a Statistical Service document published in October 2014 on the 2010 population census. I do appreciate your concern as northern Ghana is known for her high temperatures. However, during harmattan, night temperature can be even lower than 10 oC. However, due to global warming, such chilly weather is now rare.

Figures.2.1 to 2.4: Titles should read: Relationship between rain fall and leaf area expansion , etc

 Thanks. Please, this has been revised to read: Relationship between rainfall and leaf area expansion in Frafra potato, etc.

Fig. 5.2 Tuber mass. Not clear to me. I do not understand what this is about. Thanks. This has been revised to read: Fig. 2.5: Mass of tuber produced by the six varieties of Frafra potato

Other comments and author’s responses:

Abonuusum et al., 2021 not presented in reference list Thanks. Please, I am sorry for using the wrong name, Abonuusum instead of Ayimbire, which listed as number 3 in the references

Line 275 (now 262) cite plates 1, 2, 3, … Thanks. Please, this has been done.

Line 325 following: cite tables with the results and add a table showing significant differences. Thanks for drawing my attention to the need to cite the table of results and add P-Values. These have been done, I have added a row of P-Values at the bottom of table 3.2. Hope it meets your request.

CV= 56% …. CV measures experimental error and not variation…. Thanks. Please, this has been done. Have change the sentence and used P- Value instead.

Line 415 following: May be nutrient imbalance. Excessiveness in nitrogen…. Thanks very much for drawing my attention to the possibility of excessive nitrogen being the cause of the massive vegetative growth at the expense of tuber formation. However, table 3.1 clearly shows that the soil was poor in nutrient content. No chemical fertilizer was also applied. Therefore, excessive nutrient in the soil could not have been the cause of the high vegetative growth. 

Line 425: Remove Plate and number 6 Please, the statement: fig 2.1: Rainfall and leaf expansion in Frafra potato, is not a subheading. It indicates where fig 2.1 should be placed. Currently, fig 2.1 is attached as additional information after the references. Therefore, it is not clear to me when you said I should removefig 2.1, and leave the sentence hanging like that. Thank you 

Nkamah, 2004 is not in reference list This is cited in the list of references (number 14). Thank you

Lines 550 -553 Ayimbire A, Asumboya…. Not cited in text Thanks. Please, it was wrongly cited in text as Abonuusum et al., 2021, now corrected.

We note that Figure 1 in your submission contain [map/satellite] images which may be copyrighted Please, figure 1 has been replaced with a map obtained from the Ghana Statistical Service, Upper East Region. Thank you

We note that Supplementary Figure plate 4 includes an image of a participant in the study……… The signed consent form should not be submitted with the manuscript, ….. Please amend the methods section and ethics statement of the manuscript to explicitly state that the patient/participant has provided consent for publication Mr. Abdul Ganiu Anyagri Ndeogo has given written informed consent (as outlined in PLOS consent form) to publish his image in Supplementary Figure plate 4. The ethics statement has been amended. Thank you

Please upload a copy of Figure 3, to which you refer in your text on page 22 Please, it is figure 2 that was mistakenly cited as figure 3. Thank you

Please ensure that you refer to Figure 2 in your text Please, figure 2 was mistakenly cited as figure 3. This has been corrected. Thank you

One of the noted authors is a group or consortium BTU Research Group 2015. In addition to naming the author group, please list the individual authors and affiliations within this group in the acknowledgments section of your manuscript. Please, BTU Research Group 2015, is not an author of the manuscript. It is a research group in the university some of whose members are authors of this manuscript. The other members are not authors of this manuscript. Thank you.

---

## [Editor Report · Decision Letter 1]

10 Oct 2022

Comparing the Growth and Yield Performance of Six Varieties of Frafra Potato (Solenostemon rotundiflius Poir) Grown under rain-fed conditions in the Guinea Savanna Ecological Zone of Ghana

PONE-D-22-14687R1

Dear Dr. Ayimbire,

We’re pleased to inform you that your manuscript has been judged scientifically suitable for publication and will be formally accepted for publication once it meets all outstanding technical requirements.

Kind regards,

Min Huang

Academic Editor

PLOS ONE

Additional Editor Comments (optional):

The authors have addressed all comments of two reviewers. Therefore, I suggest accepting it for publication.

---

## [Editor Report · Acceptance letter]

21 Oct 2022

PONE-D-22-14687R1 

*Comparing the growth and yield performance of six different varieties of frafra potato (Solenostemon rotundifluis Poir) grown under rain-fed conditions in the Guinea Savanna ecological zone of Ghana.*

Dear Dr. Ayimbire:

I'm pleased to inform you that your manuscript has been deemed suitable for publication in PLOS ONE. Congratulations! Your manuscript is now with our production department. 

Kind regards, 

on behalf of

Dr. Min Huang 

Academic Editor

PLOS ONE